# Sub pixel resolution using spectral-spatial encoding in x-ray imaging

**Timothy P. Szczykutowicz**[1,2,3]*, **Sean D. Rose**[4], **Alexander Kitt**[5]

**1** Department of Radiology, University of Wisconsin-Madison, Madison, WI, United States of America,
**2** Department of Medical Physics, University of Wisconsin-Madison, Madison, WI, United States of America,
**3** Department of Biomedical Engineering, University of Wisconsin-Madison, Madison, WI, United States of America, **4** Department of Diagnostic and Interventional Imaging, UT Health Sciences Center at Houston, Houston, TX, United States of America, **5** EWI, Buffalo, NY, United States of America

* tszczykutowicz@uwhealth.org

**Data Availability Statement:** Data are available from the figshare database: https://figshare.com/projects/PLOS_ONE_PAPER_sub_pixel_resolution/124078.

## Abstract

### Purpose

Previous efforts at increasing spatial resolution have relied on decreasing focal spot and or detector element size. Many "super resolution" methods require physical movement of a component of the imaging system. This work describes a method for achieving spatial resolution on a scale smaller than the detector pixel without motion of the object or detector.

### Methods

We introduce a weighting of the photon energy spectrum on a length scale smaller than a single pixel using a physical filter that can be placed between the focal spot and the object, between the object and the detector, or integrated into the x-ray source or detector. We refer to the method as sub pixel encoding (SPE). We show that if one acquires multiple measurements (i.e. x-ray projections), information can be synthesized at a spatial scale defined by the spectrum modulation, not the detector element size. Specifically, if one divides a detector pixel into n sub regions, and m photon-matter interactions are present, the number of x-ray measurements needed to solve for the detector response of each sub region is mxn. We discuss realizations of SPE using multiple x-ray spectra with an energy integrating detector, a single spectra with a photon counting detector, and the single photon-matter interaction case. We demonstrate the feasibility of the approach using a simulated energy integrating detector with a detector pitch of 2 mm for 80-140 kV medical and 200-600 kV industrial applications. Phantoms used for both example SPE realization had some features only a 1 mm detector could resolve. We calculate the covariance matrix of SPE output to characterize the and noise propagation and correlation of our test examples.

### Results

The mathematical foundation of SPE is provided, with details worked out for several detector types and energy ranges. Two numerical simulations were provided to demonstrate feasibility. In both the medical and industrial simulations, some phantom features were only observable with the 1 mm and SPE synthesized 2 mm detector, while the 2 mm detector

**Funding:** The authors received no specific funding for this work.

**Competing interests:** TPS has a patent application submitted to the USPTO on the methods presented in this work. The authors have a patent application submitted to the USPTO on the methods presented in this work. There are no other patents or products in development to declare. This does not alter our adherence to PLOS ONE policies on sharing data and materials.

was not able to visualize them. Covariance matrix analysis demonstrated negative diagonal terms for both example cases.

## Conclusions

The concept of encoding object information at a length scale smaller than a single pixel element, and then retrieving that information was introduced. SPE simultaneously allows for an increase in spatial resolution and provides "dual energy" like information about the underlying photon-matter interactions.

## 1 Introduction

Fundamental trade-offs in focal spot size, detector element size, cost, and imaging time all must be considered when a high level of spatial resolution in x-ray imaging is required. Resolution is inherently limited by focal spot and detector element size [1, 2]. Many methods have been proposed to increase imaging resolution without requiring changes to the focal spot or detector element size. The field of super resolution (SR) is a method in which several low spatial resolution images are processed in order to obtain a high spatial resolution image. In the most common realizations of SR imaging, a sub pixel shift must be present between the low resolution images such that their information content is partially independent enabling a computational method to synthesize a high resolution version of the low resolution image set [3]. For many applications, the need to shift the low resolution image means physical motion of the source location, object, and/or detector array is required. Such a shift is likely the main reason we have not seen adoption of SR techniques into more imaging applications.

The present paper discusses a new approach for simultaneously obtaining x-ray line integrals of the photon-matter basis coefficients for sub regions within a detector pixel. Our approach does not require any shifting of the focal spot, object, or source. We describe a method for obtaining sub pixel spatial resolution by spatially encoding object information within a single detector element using stationary energy sensitive filters. We refer to our approach as sub pixel encoding (SPE). This paper introduces SPE and describes its mathematical foundation. Numerical simulations are provided to demonstrate feasibility, with future works characterizing SPE for specific imaging tasks.

## 2 Methods

### 2.1 Sub pixel encoding (SPE)

SPE applies a different weighting to x-ray projections within a single detector pixel. We refer to this weighting as encoding. One may divide a pixel into a number (n) of sub regions where each sub region is defined by a different energy weighting. In Fig 1 we show an example of dividing a single pixel into 2 sub regions. X-rays traversing through an imaging object and striking the detector are spatially encoded using a filter (F) with a different filter response for the left and right sides of the detector pixel. Such a filtering allows a spectral-spatial encoding of x-rays hitting the left and right sides of a single detector element. F will respond differently to different energy photons due to changes in photon-matter interaction cross sections with energy. In the sections below, we provide a theoretical explanation for how we can recover the information on the left and right side of the pixel by exploiting the spatial and energy dependent weighting F provides.

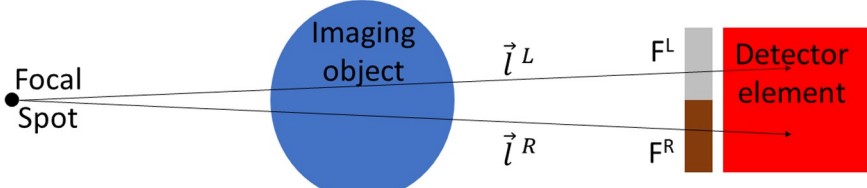

**Fig 1. Sub pixel encoding divides a single detector pixel into multiple sub regions (2 sub regions are shown in this figure).** Here we denote the 2 sub regions as the left and right sides of the detector pixel. A filter is denoted by $F$ in the figure and is split into a left $F^L$ and right $F^R$ side. X-rays emanate from the source and pass through the object and strike the left or right side of the filter and detector. We denote the paths taken by rays on the left and right sides of the detector pixel as $\vec{l}^L$ and $\vec{l}^R$ respectively.

To derive the mathematical foundation for SPE, we will first consider the case of a single measurement on a single detector pixel. The signal $S$ produced by detecting x-ray photons produced using an incident spectrum $\Omega(E)$, using a detector with a response given by $\omega(E)$, passing through an imaging object $\mu(x, y, E)$ is given by,

$$S = \int_0^{E_{max}} dE\Omega(E)\omega(E)e^{-\int d\vec{l}\mu(x,y,E)},$$ (1)

where $x$ and $y$ define spatial position in 2 dimensions, $E$ is energy, and $\vec{l}$ denotes a vector passing through $\mu(x, y, E)$ starting at the x-ray source focal spot and terminating in the detector pixel as shown in Fig 1. In reality, the signal at the detector plane will also be composed of a scattered photon component which we neglect in this derivation as did Alvarex and Macovski 1976 [4] in their seminal dual energy CT work. We can generalize an expression for x-ray measurements for different x-ray spectra and detector responses as,

$$S_{i,j} = \int_0^{E_{max}} dE\Omega(E)_i\omega(E)_j e^{-\int d\vec{l}\mu(x,y,E)}$$ (2)

where the subscript $i$ denotes the use of different incident x-ray spectra and $j$ different detector responses. For example, we can change the incident spectra by adding or removing primary beam filtration or changing the peak energy of a polychromatic x-ray spectrum [1]. Detector response may be changed, for example, by changing the detector material resulting in a different energy sensitivity, or using a detector sensitive to multiple beam energies (e.g., a photon counting or sandwich detector design) [5, 6].

SPE is enabled by applying different spectral weighting to sub regions within a single detector pixel. We accomplish this by denoting two paths from the x-ray source to the left and right sides of a detector element and having these paths intersect different halves of a filter $F$. We denote the left and right detector element vectors as $\vec{l}^L$ and $\vec{l}^R$ as shown in Fig 1. We can now describe $S_{i,j}$ as,

$$S_{i,j} = \int_0^{E_{max}} dE[F(E)^L\Omega(E)_i^L\omega(E)_j^L e^{-\int d\vec{l}^L\mu(x,y,E)} + F(E)^R\Omega(E)_i^R\omega(E)_j^R e^{-\int d\vec{l}^R\mu(x,y,E)}].$$ (3)

Here we have allowed the incident spectra and the detector response to vary for the left and right sides of the detector pixel while keeping the detector element with a single read out. The detector response could vary by using a different scintillator or semiconductor material on the left and right sides of individual detector elements (i.e., the element would still have a single readout, but regions of varying sensitivity to x-rays that would vary with energy). We have also

accounted for the presence of the filter shown in Fig 1 using $F(E)$ which takes on different values for the left and right sides of the detector.

The terms in the exponential function are given by,

$$\int d\vec{l}^{\,L} \mu(x, y, E) = PE(E) \int d\vec{l}^{\,L} A(x, y) + CE(E) \int d\vec{l}^{\,L} B(x, y) \qquad (4)$$

and

$$\int d\vec{l}^{\,R} \mu(x, y, E) = PE(E) \int d\vec{l}^{\,R} A(x, y) + CE(E) \int d\vec{l}^{\,R} B(x, y) \qquad (5)$$

where $PE(E)$ is the photoelectric basis function and $CE(E)$ is the Compton basis function [7]. $A(x, y)$ and $B(x, y)$ are the spatial distributions of the weighting terms across the imaging object for the photoelectric and Compton terms respectively [4]. For this derivation of SPE, we neglect the Rayleigh scattering contribution and also assume beam energies are below pair production energies. This derivation also neglects the presence of a material with a k edge in the energy range we acquire data within similar to Alvarez and Macovski 1976 [4]. We refer to the line integrals of the basis functions as $PAL$, $PAR$, $PBL$, and $PBR$ denoting line integrals through the imaging object's first (i.e., $A(x, y)$) and second (i.e., $B(x, y)$) basis functions weights for the left and right sides respectively.

Inspection of Eqs 3, 4 and 5 reveals that we now have 4 unknown terms in Eq 3. The 4 unknowns are the weighting coefficients for the photoelectric and Compton basis functions for the left and right side of the detector pixel. To determine these unknowns, 4 unique measurements of $S_{i,j}$ must be obtained. The following sections will describe how measurements of $S_{i,j}$ may be determined.

## 2.2 Realizing SPE using multiple measurements at different beam energy

Perhaps the most straightforward manner in which to acquire unique measurements of $S_{i,j}$ is to use different beam energies. Modulation of beam energy may be accomplished by changing the peak beam energy, changing the anode material, or filtering the beam. All of these methods are currently in use in various industrial and medical applications. Below, we provide the equations for $S_{i,j}$ acquired using different peak beam energies where index $i = 1,2,3,4$ corresponds to spectra $\Omega(E)$ of 80, 100, 120, and 140 kV respectively as

$$S_i = \int_0^{E_{max}} dE \Omega(E)_i \omega(E) [F(E)^L e^{-\int d\vec{l}^{\,L} \mu(x,y,E)} + F(E)^R e^{-\int d\vec{l}^{\,R} \mu(x,y,E)}, \qquad (6)$$

where we have kept the detector response $\omega(E)_j$ unchanged for all measurements and $E_{max}$ is the maximum energy present in $\Omega(E)_i$. The measurements shown in Eq 6 represent 4 unique realizations of $S$ suitable for determining the 4 unknowns in the SPE derivation ($m = 2$, $n = 2$) from Section 2.1.

## 2.3 Realizing SPE using an energy sensitive detector

In place of using 4 different beam energies, one could use a detector with an energy sensitive response $\omega(E)_j$. If 4 different detector sensitivities are measured, we can collect 4 unique measurements of $S_{i,j}$ suitable for SPE in the $m = 2$, $n = 2$ energy and spatial resolution condition. Such a detector sensitivity is realizable, for example, using a photon counting detector and

would provide a $S_j$ measurement for each of j = 1,2,3,4 energy thresholds as,

$$S_j = \int_0^{E_{max}} dE\Omega(E)[F(E)^L\omega(E)_j e^{-\int \vec{dl}^L\mu(x,y,E)} + F(E)^R\omega(E)_j e^{-\int \vec{dl}^R\mu(x,y,E)}] \qquad (7)$$

where we have kept the incident imaging spectrum $\Omega(E)_i$ fixed for a single irradiation of the energy sensitive detector.

## 2.4 General considerations for SPE

We can appreciate from Eqs 3, 4 and 5 that for each division of a detector pixel into n regions, n unique measurements times the number of basis functions m are required to obtain the line integrals of the basis function coefficients. In Sections 2.2 and 2.3 we assumed a polychromatic spectra was used with photons in the kV energy range. This means two basis functions are required to realize the SPE method representing the photoelectric and Compton interactions requiring 4 (n = 2, m = 2) measurement to obtain a halving of detector element size. For cases in which higher beam energies are used like with higher kV (i.e., hundreds of kV) voltage x-rays used in industrial imaging Compton scattering dominates for many materials. In such cases where only one photon-matter interaction is present, the data uniqueness requirements would be reduced, with only 2 (n = 2, m = 1) unique measurements needed. Assumptions on how many photo-matter interactions "dominate" at a specific energy for determining m will also be a function of material because of differing Photoelectric contribution with atomic number [7]. For energy levels over the pair production threshold, additional cross sections would need to be added increasing the number of required measurements.

## 2.5 Experimental validation

**2.5.1 Medical imaging case.** We demonstrate SPE using a simulation of 2-d projection (i.e., radiograph) x-ray imaging. We simulated a 100 mm x 100 mm imaging object of varying thickness. The object thickness was composed of 20 mm of water with a varying thickness of bone. The bone thickness was set to decrease linearly from 1 to 0 mm over 25 mm, then three regions of 25 mm each were modulated using the absolute value of a sinusoidal function having periods of 50, 25, and 4 mm respectively. The bone sinusoid had a magnitude of 1 mm. The object is shown in Fig 2. A polychromatic spectra was simulated at 70, 100, 120, and 150 kV with an inherent filtration amount of 2.5 mm of Al and 500 mAs [8]. The spectrum generator uses modeled and measured data to provide realistic X-ray spectra for the simulated techniques and inherent filtration. The projection imaging was simulated using a parallel beam geometry with 2x1 mm and 1x1 mm detectors. The function $F$ was constructed by simulating a 0.2 mm thickness of $HgI_2$ across one side of each detector element. The filter was used only for the 2 mm detector, the ideal detector data was filterless. We modelled the detector signals using the formalism provided in Eq 6 (i.e., energy integrating detectors with a linear energy response and perfect detection efficiency). For basis functions, we used water and bone in place of the photoelectric and Compton basis functions from Eq 6. A numerical equation solver (vpasolve, Mathworks Natwick MA) was used to solve for the projection line integrals of $A(x, y)$ and $B(x, y)$ for the left and right sides of each detector pixel.

We used the basis coefficients and known energy response of the basis functions [9] to reconstruct 60 keV monochromatic SPE enabled line profiles, $SPE^L$ and $SPE^R$ at 60 keV where $SPE^L = PAL \cdot \mu_{basis1}(E)|_{E=60keV} + PBL \cdot \mu_{basis2}(E)|_{E=60keV}$ and $SPE^R = PAR \cdot \mu_{basis1}(E)|_{E=60keV} + PBR \cdot \mu_{basis2}(E)|_{E=60keV}$. These SPE profiles correspond to the monochromatic line integrals of the left and right sides of the detector. We can produce a similar, albeit polychromatic version of the SPE derived monochromatic profiles from the single

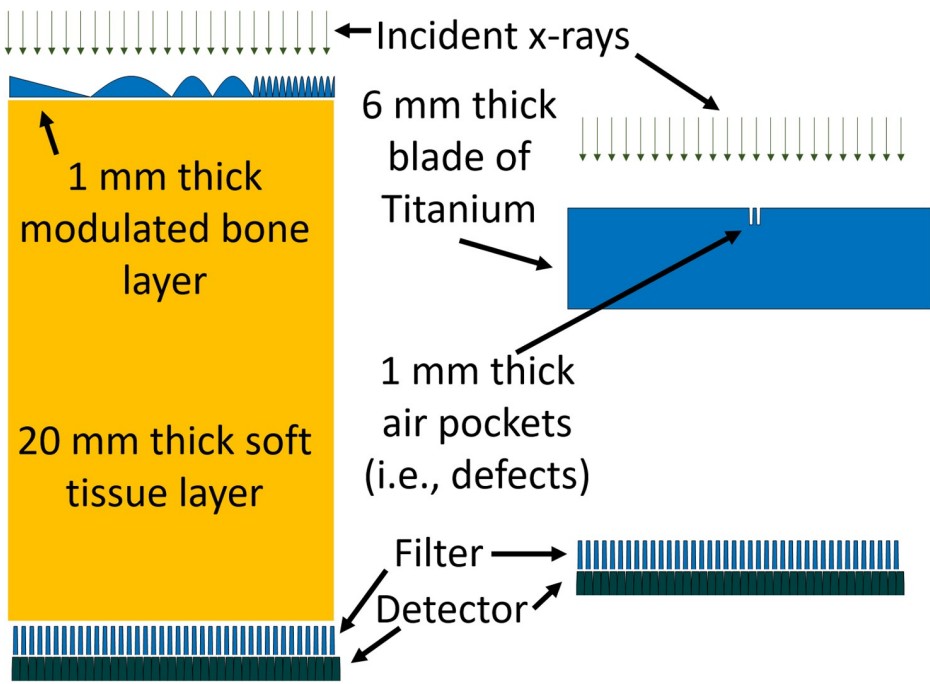

**Fig 2. The medical and industrial simulation examples are shown on the left and right of this figure respectively.** Figure not drawn to scale.

energy data. To do so, the single energy detector measurements were transformed into projection line integrals by log transforming the detector signal with no phantom in place by the measurements $S$ resulting in an effective energy representation of the projection line integral (e.g., at 80 kV $ln \frac{\int_0^{80kV} dE\Omega(E)_{80\ kV}\omega(E)[F(E)^L + F(E)^R]}{S_{80\ kV}}$).

**2.5.2 Industrial imaging case.** A second simulation was performed using an industrial imaging task. A 10 cm long, 1 cm wide, and 6 mm thick piece of titanium was imaged through the 6 mm dimension. Two defects were placed near the center of the titanium phantom. Each defect was 1 cubic mm in size and the two defects were placed 1 mm apart along the long dimension of the titanium phantom. The object is shown in Fig 2. A 200 and a 600 kV spectra were created using the TASMICS spectra generator [10]. The spectrum generator uses modeled and measured data to provide realistic X-ray spectra for the simulated techniques and inherent filtration. Each specta was scaled to deliver1 mGy in air and then filtered by 3 mm of Sn. The filter F was constructed from 5 mm of Mo on one side of the detector element and zero thickness on the other side. The filter was used only for the 2 mm detector, the ideal detector data was filterless. We modelled the detector signals using the formalism provided in Eq 6 (i.e., energy integrating detectors with a linear energy response and perfect detection efficiency). We used a single basis function of Titanium. Using a single basis function to model the attenuation assumes only 1 photon matter interaction is present, which for this energy range we ignored the Photoelectric effect and assumed our signal was dominated by Compton Scattering. A numerical equation solver (vpasolve, Mathworks Natwick MA) was used to solve for the projection line integrals $B(x, y)$ for the left and right sides of each detector pixel. Similar to the medical case described previously, we also created monochromatic SPE enabled projection measurements at energies of 145.5 keV and 270.5 keV set to match the attenuation of the polychromatic log normalized measurements acquired at 200 and 600 kV.

We compute the standard deviation over a uniform region of Titanium thickness (3 cm by 1 cm wide) projection measurement for the 200 kV 1 mm detector, the 200 kV 2 mm detector, and the 145.5 keV SPE monochromatic projection to compare the noise between the acquisitions.

## 2.6 Covariance matrix analysis

To characterize noise propagation properties of SPE, we calculated the covariance matrix for the problem instances of SPE described in Sections 2.5.1 and 2.5.2. A system with no correlation in noise would have all off diagonal elements equal to zero. The noise variance is equal to the diagonal terms of the co-variance matrix, such that the signal to noise (SNR) can be obtained using the diagonal covariance matrix elements. SNR calculated in this way, however, ignores noise correlations which are know to effect observer performance on detection tasks. We will report the SNR using the square root of the diagonal covariance matrix entires for the medical and industrial cases. Negative covariance matrix entries indicate cases where when one output increases due to noise, another output tends to decrease. For a more extensive discussion on noise, noise correlation, and observer performance with different types of noise, see Reference [11] A fair characterization of the noise and signal detection properties of SPE compared to other methods would require optimization of the thickness and material for F for specific problems in medical and industrial imaging. The purpose of the analysis explained in this section was simply to demonstrate the presence or absence of off-diagonal terms and the polarity of the covariance matrix entries. From Section 2.5.1 we modelled 20 mm of water over the left and right sides of a single detector element and 1 mm of bone over 1 side of the detector element. From Section 2.5.2 we modelled 6 mm of Titanium over the left and 5 mm of Titanium over the right side of a single detector element. The first order approximation of the covariance matrix is given by,

$$\mathbf{K}_p \approx \mathbf{J}_S^{-1} \, \mathbf{K}_S (\mathbf{J}_S^{-1})^T \tag{8}$$

where $\mathbf{K}_p$ is the covariance matrix of the SPE outputs (i.e., f($\mathbf{S}$) = [PAL PAR PBL PBR] for the medical case and f($\mathbf{S}$) = [PBL PBR] for the industrial case), $\mathbf{J}_S^{-1}$ is the inverse of the Jacobian of the forward model, and $\mathbf{K}_S$ is the covariance matrix of the input measurements $\mathbf{S}$ which we assume to be diagonal. The general form of $\mathbf{S}$ was given in Eq 3, and implementation specific forms of $\mathbf{S}$ in Eq 6 for the case when SPE is realized with different beam energies and in Eq 7 with an energy sensitive detector. We experimentally determine $\mathbf{K}_S$ by running 1,000 instances of $\mathbf{S}$ adding Poisson noise to the raw detector measurements [11]. The function *f* is fully specified and we can calculate its Jacobian, but we do not show it here as it has 16 entries for the medical imaging case 8 entries for the industrial imaging case. Here we have used the inverse function theorem which says that the inverse of the Jacobian of the forward model is equal to the Jacobian of the function that maps $\mathbf{S}$ to f($\mathbf{S}$) assuming $J^{-1}$ and its inverse are continuously differentiable in the neighborhood of our problem instance.

## 3 Results

Figs 3 and 4 depict 2D detector images and line profiles of the 1x1 mm, 1x2 mm, and various SPE derived images for the medical imaging case. The 4 mm period bone modulation is not visible on the 1x2 mm detector images or profiles. Images acquired using the 1x1 mm detector do resolve the 4 mm period bone pattern. Using data acquired only from the 1x2 mm detector, the SPE derived images do enable the 4 mm bone pattern to be resolved as seen in Figs 3c and 4c–4e.

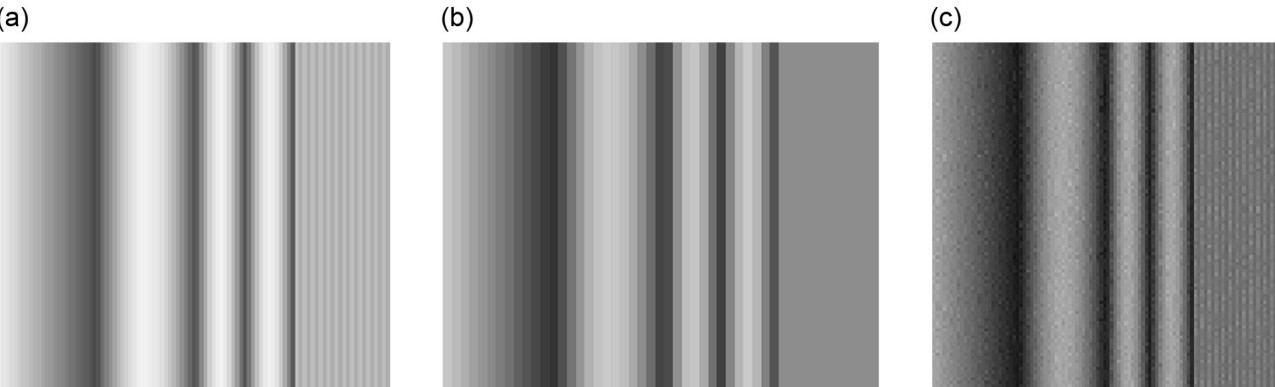

**Fig 3.** (a) depicts the log transformed detector signal (i.e., projection integral) from the 1x1 mm simulated detector element at 120 kV. (b) depicts the log transformed detector signal (i.e., projection integral) from the 1x2 mm simulated detector element at 120 kV. (c) depicts the SPE derived 60 keV projection line integral reconstructed using data only from a 1x2 mm detector. Pay attention to the right side of the images, the high frequency bone modulation pattern is only visible on the 1 mm and 1 mm SPE derived images. Images are unitless arguments to an exponential function (i.e., projection line integrals) and are displayed at a window setting of [0.4 0.51].

Figs 5 and 6 depict 2D detector images and line profiles of the 1x1 mm, 1x2 mm, and various SPE derived images for the industrial imaging case. The pore voids are not visible as being 2 distinct defects on the 1x2 image. Images acquired using the 1x1 mm detector do resolve the pore voids as two separate voids. Using data acquired only from the 1x2 mm detector, the SPE derived images do enable the pore void defects to be individually resolved. Line profiles in Fig 6 also confirm that only the 1x1 detector and image contrast types derived from SPE allow the pore void defects to be individually resolved.

For the medical example, the covariance matrix was

$$\mathbf{K}_p = \begin{bmatrix} 5.4516 & -3.3601 & -0.9957 & -0.6758 \\ -3.3601 & 2.4719 & 0.6473 & 0.1896 \\ -0.9957 & 0.6473 & 0.1849 & 0.1039 \\ -0.6758 & 0.1896 & 0.1039 & 0.2132 \end{bmatrix} mm^2.$$

For the industrial example, the covariance matrix was

$$\mathbf{K}_p = \begin{bmatrix} 0.0035 & -0.0084 \\ -0.0084 & 0.0230 \end{bmatrix} mm^2.$$

Both the medical and industrial examples have non-zero off-diagonal terms and negative terms. The SNR for the medical case were: for 20 mm water thickness on the un-filtered detector side $20/\sqrt{(5.45)} = 8.57$, for 20 mm water thickness on the right filtered detector side $20/\sqrt{2.47} = 12.73$, for 1 mm of bone on the unfiltered detector side $1/\sqrt{0.18} = 2.32$, and for 1 mm of bone on the filtered detector side $1/\sqrt{0.21} = 2.16$. The SNR for the industrial case were: for the 6 mm of Titanium on the unfiltered detector side $6/\sqrt{0.0035} = 101.42$ and for the 5 mm of Titanium on the filtered detector side $5/\sqrt{0.02} = 32.97$

The noise (i.e., standard deviation of projection values over a uniform region) of the industrial imaging case, taken from projection images acquired at 200 kV for the 1 and 2 mm

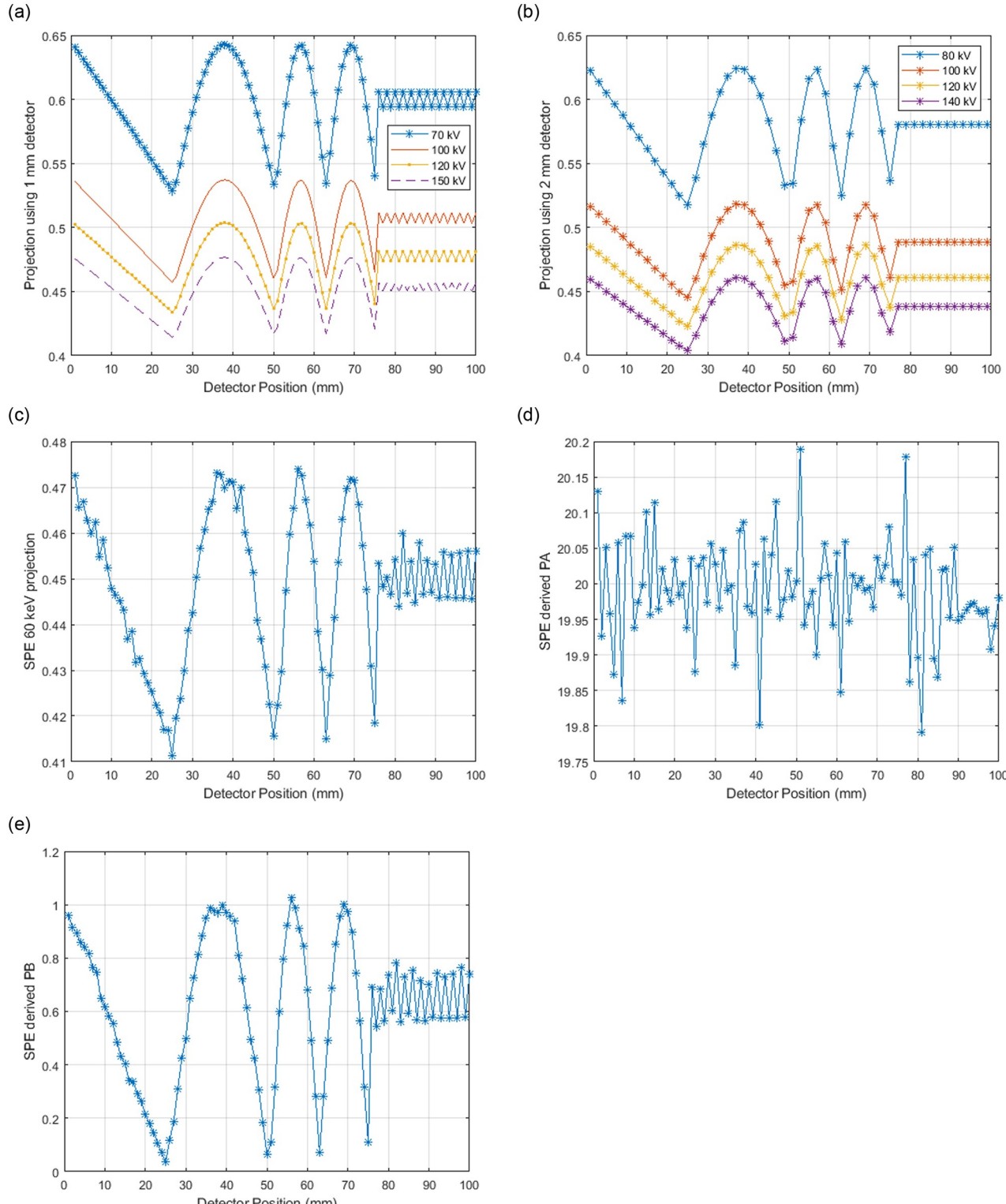

**Fig 4.** 1 mm detector element line profile are shown in (a), 2 mm detector element line profiles are shown in (b), a SPE derived 60 keV line profile with a synthesized resolution of 1 mm detector elements is shown in (c). (d) is the contribution from PA representing the water contribution and (e) is the contribution from PB representing the bone contribution.

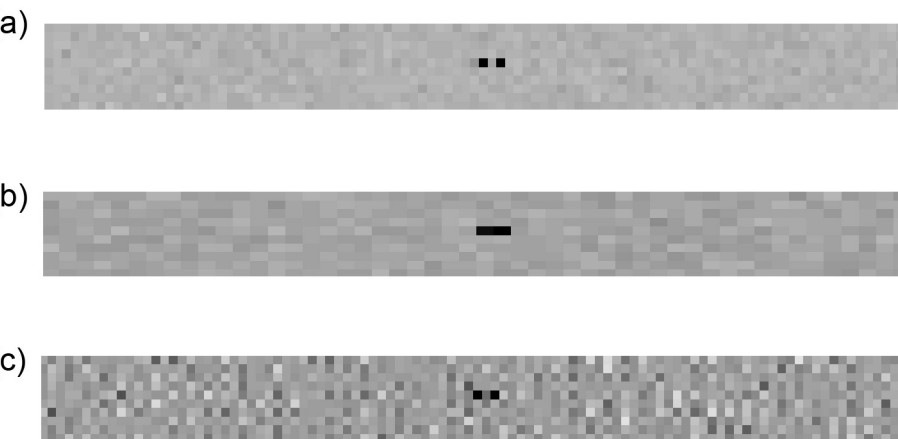

**Fig 5.** Projection line integral images of the Titanium blade with 1 mm³ air pocket defects imaged using (a) 1 mm detector elements, (b) 2 mm detector elements, and (c) 2 mm detector elements using SPE. The defect is only correctly shown as two distinct air pockets in the 1 mm and 1 mm SP derived images. Images are unitless arguments to an exponential function (i.e., projection line integrals) and are displayed at a window setting of [0.4 0.5].

detector cases and 145.5 keV for the SPE derived 1 mm projection image were 0.0023, 0.0026, and 0.0083 (projection images are unitless) respectively.

## 4 Discussion

In this paper we defined a new method for increasing the spatial resolution of an x-ray imaging system while simultaneously enabling determination of the contribution of each photon-matter basis interaction. In short, we demonstrated a mathematical foundation and demonstrated how SPE allows "super resolution" while providing a "dual energy" like data acquisition. The basis plots of Fig 4 show how the SPE derived projections not only allow detection of the 4 mm period bone signal (e.g., Fig 4e), but also accurate decomposition of the projection data into the soft tissue thickness of 20 mm as shown in Fig 4d. The 1x1 mm detector was also able to resolve the 4 mm period bone signal, but did not allow decomposition into bone and soft tissue components. Based on the noise amplification observed in Fig 3 (e.g., between Fig 3a and 3c) and Fig 4 (e.g., between Fig 4a and 4c–4e) it is possible that implementation of SPE would be clinically unacceptable using the example conditions used in this paper. The industrial results shown in Figs 5 and 6, however, did show immediate promise as the noise amplification for the SPE results was not as severe relative to the 1x1 mm detector example as it was for the medical example.

To combat the noise increases we observed, it is likely that SPE will benefit from a denoising stratagem that takes advantage of the covariance matrix having negative and off diagonal terms and or constraining the SPE result by the lower noise original S measurements [12, 13]. Future work should be done to consider these possibilities. For both the medical and industrial cases the SNR varied across the left and right sides of the detector element, with the medical case also exhibiting different SNR for the water and bone signals. More work needs to be done on specific classes of problem instances for medical and industrial cases before we can comment on any general trends about comparing the SNR behind the unfiltered and filtered halves on the detector.

For the industrial imaging case, the noise was slightly higher for the 2 mm projection relative to the 1 mm because the filter was used for the 2 mm projection data but not the 1 mm "ideal" detector imaging. The noise increase for the SPE compared to 1 mm detector was

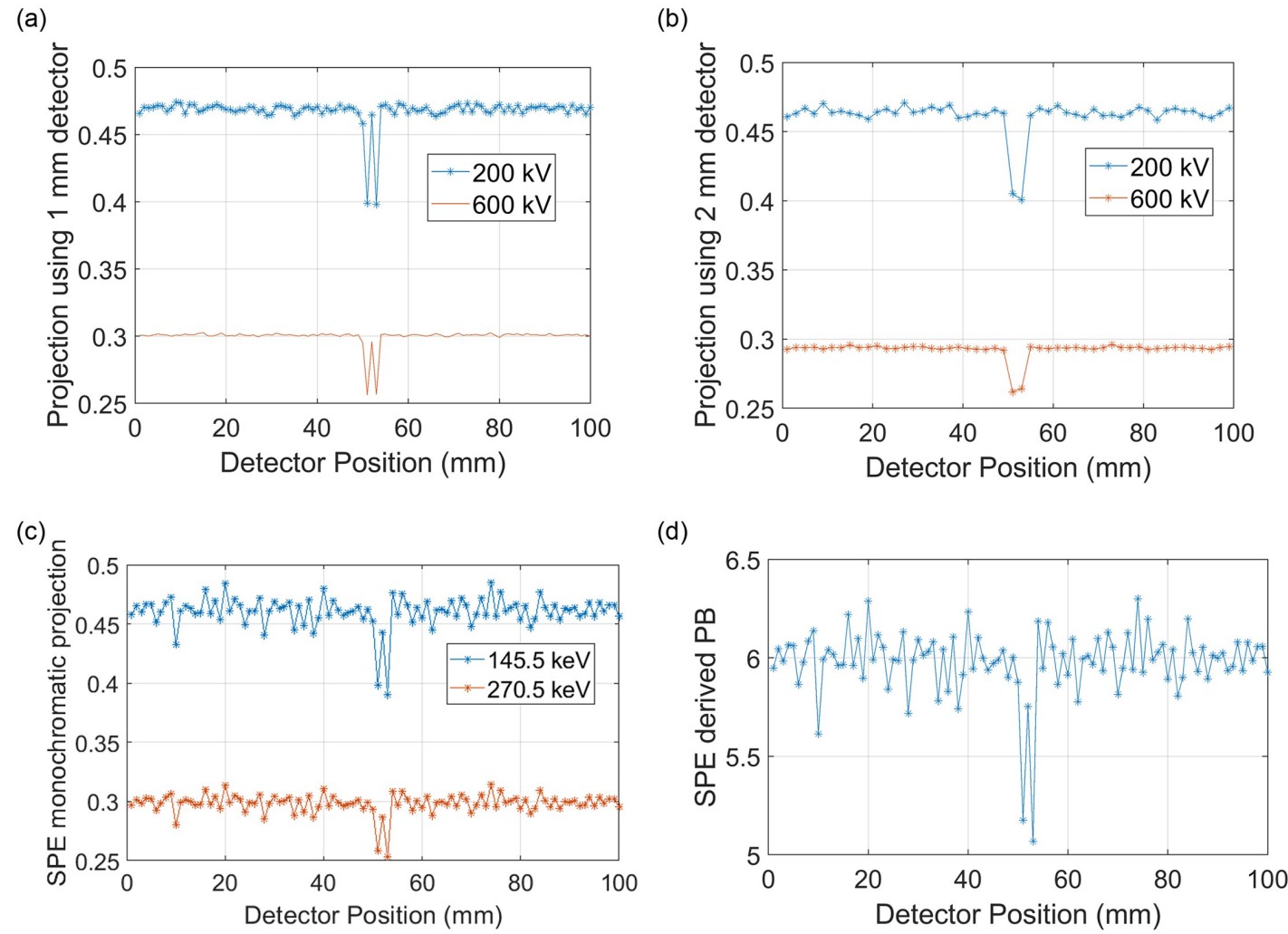

**Fig 6.** Line profiles for the 1 mm, 2 mm and 2 mm detector using SPE in (a), (b), and (c) respectively. (d) shows the PB line integral derived from the 2 mm detector element data using SPE.

0.0083/0.0023 = 3.6 times. The SPE derived 1 mm projection in this comparison used more photons than the single ideal detector comparison. This is because the SPE derived projection used 2 projection images, at 200 and 600 kV from the 2 mm detector to create the 145.5 mm SPE projection image which we compared to a single projection image from the 1 mm detector at 200 kV. This comparison was not meant to fully characterization the noise amplification of SPE and future works should address specific imaging cases where imaging time and or ionizing radiation dose are kept constant. These initial results seem promising, as there are likely many applications which could benefit from obtaining a 2 times resolution gain requiring only a minor hardware modification (i.e., introducing the filter F in front of the detector plane) in stead of the cost of a new detector with double the detector ASIC hardware and smaller detector elements.

We assumed a physical filter F was placed directly before the detector as this location would likely be the easiest to experimentally implement. This implementation of SPE would most closely mimic retrofitting existing x-ray imaging devices for SPE data acquisition; one just places an absorbing filter over their detector with filters aligned over sub regions within

individual detector elements. As the F filter becomes so thick as to block the the detector signal to part of the detector completely, SPE approaches a "high resolution comb" like spatial resolution increasing solution [14]. Preliminary studies by our group have shown that as the filter material is changed to higher atomic number or made thicker on one side versus the other, stronger changes in filter value are obtained with energy. This is good for providing the independent measurements required by SPE on the different regions of the detector. Such conditions, however, increase the attenuation of the signal if the filter is placed between the object and the detector read out which increases noise. So careful optimization of the filter material and thickness and placement is required in future studies of the SPE method. Placing F on the focal spot side of the imaging system is also possible as such a placement would not change the mathematical foundation for SPE. Albeit, placing F close to the focal spot may degrade the spatial encoding on the detector due to focal spot blurring. It may be possible to also encode F directly into the x-ray generation, perhaps using an anode made of different materials over the length of the focal spot, cold x-ray emission at different energies, or a energy selective separation of the beam as is common using brilliant sources. Integration of *F* into the detector could be achieved by varying the sensitive detector material at a higher frequency than the detector read out electronics allowing for encoding at a length scale smaller than the spatial detector read out interval.

For industrial applications, where dose is not a concern, SPE may allow for cheaper x-ray imaging systems to be developed by increasing the detector element size and maintaining spatial resolution. Albeit, our work has demonstrated noise will increase for SPE relative to using a "true" higher resolution detector and offsetting this increase via longer scan times is not desirable for many industrial applications [15]. Implementation at industrial x-ray imaging energies (i.e., hundreds of kV) only requires two measurements as the dominate photon-matter interaction is only the Compton cross section. A possible application of SPE for industrial imaging would be visualization of small pores on big parts. For example, consider this example radiographic (i.e., 2 dimensional imaging) industrial imaging case: a 2kx2k pixel detector, a 400x400 mm part, and a 300 micron expected defect size (i.e., pore), and the condition that we need 3 voxels per defect. This means we need 100 micron resolution giving you a total image size of 200x200 mm. This would require 4 scans with movement of the part and or the imaging system between each scan. With SPE, we could use a magnification allowing our detector to span 400x400 mm at the needed 100 micron spatial resolution and never have to move the part and or imaging system, albeit we would need to acquire images at 4 (n = 4, m = 1) energies or use a photon counting detector and a single measurement. Similarly, for a tomographic non destructive evaluation of a 400x400x400 mm part with the same imaging task, 8 region of interest scans would be required without SPE versus 1 scan position for SPE.

As stated in the introduction, other methods for sub pixel resolution exist. These methods are usually known in the literature by the name "super resolution" [3]. The drawback of these techniques is that they usually require multiple exposures combined with physical movement of the focal spot, object, or detector array. SPE as implemented in this paper does not require any physical movement. We did show example implementations of SPE that used multiple exposures at different peak beam energies, however use of a spectral resolving detector would have mitigated the need for multiple exposures. Current solutions for sub pixel resolution have been commercialized and also rely on multiple measurements using super resolution sub detector pixel shifting [16] and actual are reported to increase signal to noise ration while simultaneously improving spatial resolution. We expect our solution to allow for similar gains in detection of small details, as some signal detection tasks require high spatial resolution. As we demonstrated in this paper, some detection tasks are not limited by noise, but by spatial resolution. Compared to current solutions for sub pixel

resolution [16], SPE mitigates the need for detector array shifting. SPE does allow data acquisition in a single measurement if the detector has spectral resolving capabilities. Since SPE couples the energy and spatial resolution properties of a system together, we expect SPE to behave in a non-linear fashion and require application specific optimization. SPE therefore may be best suited for applications in industrial non-destructive testing where the object size and composition are known *a prori*.

In some ways, SPE is similar to color filter arrays (CFA) and demosaicking processing [17]. CFA places a color filter over individual detector elements in visible light photography. CFA filter placement reduces the color signal spatial resolution but enables a spectrally sensitive detector. SPE is like the "inverse" of CFA, since we are placing the filters within a single detector element and our goal is not spectral sensitivity but improved spatial resolution. The "chessboard" approach used by some photon counting systems is similar to SPE in that different sub regions of the detector are sensitive to different energy ranges based on differing energy threshold ranges over the surface of the detector [18]. Albeit, in the "chessboard" approach, each region provides a unique readout unlike with SPE where 1 readout is shared over multiple energy sensitive regions.

Future work will include studying the conditioning and noise properties of SPE for imaging tasks. More work remains to specifically evaluate optimal (1) inherent beam filtration type and thickness, (2) filter F material and thicknesses, (3) energy thresholds if an energy sensitive detector is used, and (4) beam spectra energy. Due to the non linear nature of SPE, system conditions will likely not be optimal for all measurement conditions, so application specific optimization is warranted. Metrics for such an analysis must consider the spatial correlated noise and non-linearities inherent to SPE. So if tradiational image quality metrics are used (e.g., MTF and NPS), they should be titrated through ionizing radiation dose, phantom material and size, and object size and shap. The same titrations would be required for model based metrics, due to the correlated noise and non linear equations that need to be solved to realize SPE. Similar to current issues with spectral resolving detectors (i.e., photon counting), SPE would suffer from "k-escape" and signal leakage via other scattering events if F was placed directly in front of or integrated into the detector itself [5]. Evaluation of these types of energy and spatial resolution degrading phenomena would also warrant further study. According to Trefethen and Bau 1997, "a problem is well-conditioned if its relative conditioning number ($\kappa$) is small (e.g., 1, 10, $10^2$), and ill-conditioned if $\kappa$ is large (e.g., $10^6$, $10^{16}$)" [19]. While not shown in this work, our preliminary conditioning analysis demonstrates our problem has $\kappa$ values in the 150-180 range for the medical and industrial problem instance examples shown here. Future work will further explore problem conditioning, stability, and more noise covariance analysis in relation to existing imaging methods.

## 5 Conclusion

We introduced a method for encoding and then retrieving spatial object information at a length scale smaller than a single pixel element using spectral information. SPE yields an x-ray projection measurements for each photon-matter interaction for each of the sub regions a pixel is "spectrally" encoded into. Therefore, SPE simultaneously allows for an increase in spatial resolution and provides "dual energy" like information about the underlying photon-matter interactions. Our simulations demonstrated SPE is feasible in both medical and industrial applications, albeit scan time and dose efficiency remains to be characterized for specific applications. Our simulations also demonstrated noise amplification which may keep SPE from being applied to diagnostic medical imaging. SPE could be a promising method to enable cost effective increases in spatial resolution for higher kV industrial imaging and both spatial

resolution and "dual-energy like" imaging at energy ranges where multiple photon-matter interactions are present.

## Author Contributions

**Conceptualization:** Timothy P. Szczykutowicz.

**Formal analysis:** Timothy P. Szczykutowicz, Sean D. Rose, Alexander Kitt.

**Methodology:** Timothy P. Szczykutowicz, Sean D. Rose, Alexander Kitt.

**Project administration:** Timothy P. Szczykutowicz.

**Resources:** Timothy P. Szczykutowicz, Sean D. Rose, Alexander Kitt.

**Software:** Timothy P. Szczykutowicz.

**Supervision:** Timothy P. Szczykutowicz.

**Validation:** Timothy P. Szczykutowicz, Sean D. Rose.

**Visualization:** Timothy P. Szczykutowicz, Sean D. Rose, Alexander Kitt.

**Writing – original draft:** Timothy P. Szczykutowicz.

**Writing – review & editing:** Timothy P. Szczykutowicz, Sean D. Rose, Alexander Kitt.

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
