## [Decision Letter · Decision Letter 0]

11 Jun 2021

PONE-D-21-12689

Sub pixel resolution using spectral-spatial encoding in x-ray imaging

PLOS ONE

Dear Dr. Szczykutowicz,

Thank you for submitting your manuscript to PLOS ONE. After careful consideration, we feel the manuscript presents a new and potentially interesting method for improving spatial resolution, but the current presentation of data does not support the conclusion that spatial resolution was improved. Therefore, we invite you to submit a revised version of the manuscript that addresses the points raised by each reviewer.

Most critically, both reviewers found support for the conclusion of sub-pixel resolution to be weak or questionable, due to a lack of quantitative assessment and limitations in the simulations.  Reviewer 1 also requested a more quantitative assessment of image noise. Both reviewers also highlighted that results for a photon-counting detector would provide greater support for the conclusion.  Both reviewers suggested a number helpful clarifications and improvements in the manuscript presentation.

The editor's assessment is that the author's deserve an opportunity to overcome these major concerns, but it may not be possible to convincingly demonstrate sub-pixel resolution rather than merely improvements in image contrast due to basis material decomposition, which is already accomplished in dual-energy CT.

io to enhance the reproducibility of your results. Protocols.io assigns your protocol its own identifier (DOI) so that it can be cited independently in the future. For instructions see: http://journals.plos.org/plosone/s/submission-guidelines#loc-laboratory-protocols . Additionally, PLOS ONE offers an option for publishing peer-reviewed Lab Protocol articles, which describe protocols hosted on protocols.io. Read more information on sharing protocols at https://plos.org/protocols?utm_medium=editorial-email&utm_source=authorletters&utm_campaign=protocols .

We look forward to receiving your revised manuscript.

Kind regards,

Ryan K. Roeder, PhD

Academic Editor

PLOS ONE

2. Please update your submission to use the PLOS LaTeX template. The template and more information on our requirements for LaTeX submissions can be found at http://journals.plos.org/plosone/s/latex .

“TPS has a patent application submitted to the USPTO on the methods pre sented in this work. AK has signed a NDA with UW Madison to work on this project, but no license fee or fee of any kind was exchanged for during the course of this work. Data was simulated using Matlab and the TASMICS x-ray generator. Source code for implementing the results of the paper will be hosted online once this paper is accepted via a link from https://radiology.wisc.edu/profile/timszczykutowicz”

“The authors received no specific funding for this work.”

4. We note that you have a patent relating to material pertinent to this article. Please provide an amended statement of Competing Interests to declare this patent (with details including name and number), along with any other relevant declarations relating to employment, consultancy, patents, products in development or modified products etc. Please confirm that this does not alter your adherence to all PLOS ONE policies on sharing data and materials, as detailed online in our guide for authors http://journals.plos.org/plosone/s/competing-interests by including the following statement: "This does not alter our adherence to  PLOS ONE policies on sharing data and materials.” If there are restrictions on sharing of data and/or materials, please state these. Please note that we cannot proceed with consideration of your article until this information has been declared.

Additional Editor Comments (if provided):

Reviewers' comments:

Reviewer's Responses to Questions

**Comments to the Author**

1. Is the manuscript technically sound, and do the data support the conclusions?

Reviewer #1: Yes

Reviewer #2: Partly

2. Has the statistical analysis been performed appropriately and rigorously? 

Reviewer #1: N/A

Reviewer #2: Yes

3. Have the authors made all data underlying the findings in their manuscript fully available?

Reviewer #1: Yes

Reviewer #2: No

4. Is the manuscript presented in an intelligible fashion and written in standard English?

Reviewer #1: Yes

Reviewer #2: Yes

5. Review Comments to the Author

Reviewer #1: The authors in this manuscript describe a spatial-spectral sub-pixel encoding (SPE) scheme to achieve super resolution. The idea is interesting, and as demonstrated by the authors, has clear benefits, specifically for industrial applications where radiation dose is not a concern. I have the following suggestions to the authors.

The points about noise amplification appear to be qualitatively described. The authors should consider describing the noise amplification objectively, using measurements from the original 2 mm image, 1 mm image and 2 mm SPE image. This will give the readers an idea as to how strong the noise (amplification) is, to emphasize the importance of noise reduction strategies.

There is a description for realizing SPE using photon-counting detectors (2.3), however the simulations and results are for energy-integrating detector. Can the authors provide a simple example simulation for a photon-counting detector setup? This could strengthen the manuscript.

Page 7, Lines 114 to 116: "Here we have allowed the incident spectra and the detector response to vary for the left and right sides of the detector pixel. We have also accounted for the presence of the filter shown in Figure 1 using F(E) which takes on different values for the left and right sides of the detector." I have trouble understanding this statement. If I understood the SPE correctly, the detector response for left and right sides of the pixel would be identical, considering that the actual pixel (with two defined sub-pixel regions) is a single unit, therefore has a single response instead of two? The variation in response is introduced in F(E) due to the filters, if this is what the authors are referring to. Please rephrase and clarify.

Figure 2: For a general reader, it is hard to locate the 4 mm bone pattern referred by the authors. I suggest adding arrow marks to point to the regions of interest, and possibly include zoomed-in inset images. Also, it would help to include a schematic of the phantom design.

Figure 4: Window setting is [0.4 0.5], please state the unit for the image intensity value.

The line profiles in Figure 3 are a semi-quantitative way to show spatial resolution improvements. I request the authors to consider reporting MTF if possible, using the slanted-edge technique (based on edge spread function). This is a standard method widely accepted in the imaging community to quantify spatial resolution for new systems or techniques. This is another addition that I feel can strengthen this manuscript, and request the authors to consider this metric.

Reviewer #2: This is a review of "Sub pixel resolution using spectral-spacial encoding in x-ray imaging". The authors present a new technique named sub pixel encoding (SPE) to achieve super resolution in imaging. The authors present arguments where this technique is superior to previous methods where the object and/or imaging system components do not have to be physically moved.

The major concern with the manuscript is the authors have not convincingly demonstrated that spatial resolution has been improved. Rather, the approach and data would support a new method for basis material decomposition which is more likely enabling the ability to resolving objects. While this work was demonstrated through simulations, it is not apparent this would actually work on an imaging system with energy integrating detectors where there wouldn't be separate readouts for the described left and right filters. The authors do not address this but weighting factors could be used as an estimate which is done in some dual-energy CT applications. However, this would require a prior knowledge and potentially limit the application in medical imaging.

Additional comments

1. There isn't a discussion on the choice of filter material along with potential other materials that could be considered.

2. It is not clear in the methods if the fluence rate was adjusted when kV changes were applied.

3. It is not clear in the methods how the detector response was modeled for different energies.

4. Figure 2 would be improved if an image of the digital reference object was presented. It is difficult to appreciate what was being imaged.

5. The ability to resolve the defects using SPE in Figure 4(c) is not very convincing. An observer would have to have more information for interpretation.

6. The discussion on color filter arrays and demosaicking processing is confusion. Color filter arrays allow selected visible light of a particular energy and corresponding frequency to transmit. It is not clear if the point is higher frequency photons are transmitted to provide higher spatial resolution. This would likely only be applicable to high contrast objects and would degrade image contrast, thus severely limiting application.

The theoretical basis and statistical approach for this work is reasonable. The data and results simply don't support the conclusions.

6. PLOS authors have the option to publish the peer review history of their article (what does this mean?). If published, this will include your full peer review and any attached files.

Reviewer #1: No

Reviewer #2: No

---

## [Author Response · Author response to Decision Letter 0]

12 Jul 2021

I uploaded a word doc for this, thanks.

---

## [Decision Letter · Decision Letter 1]

12 Aug 2021

PONE-D-21-12689R1

Sub pixel resolution using spectral-spatial encoding in x-ray imaging

PLOS ONE

Dear Dr. Szczykutowicz,

Thank you for resubmitting your revised manuscript to PLOS ONE. Reviewer 2 declined to review the revision.  A new reviewer (3) was solicited whose expertise is more in line with your request that the paper be considered on the merits of the a new methods and its mathematical foundation.  Reviewers 1 and 3 agreed that the method has merit to be published as such but raised points that must be addressed prior to publication (see responses to PLOS ONE criteria 2 and 3 below). Therefore, we invite you to submit a revised version of the manuscript that addresses the points raised by the reviewers.

Specifically, in comments directed to the editor, Reviewer 1 agreed that MTF and advanced quantitative metrics for image quality are too big an ask for this foundational paper, but maintains that frequent use of "noise amplification" throughout the manuscript begs for some kind of quantitative support. Reviewer 1 requests a "simple" ROI-based quantitative description of noise amplification be provided, which can include acknowledging the current suboptimal simulation parameters. Reviewer 1 further suggested the authors provide a brief supporting discussion about optimizing simulations for future work to include advanced metrics such as MTF, NPS, detectability etc. 

Similarly, Reviewer 3 requested that issues surrounding the presented "covariance" should be presented along with their implications. Reviewer 3 provided a number of suggestions in this regard but communicated that his/her intent is not to demand a precise statistical analysis but necessary information to support the results and acknowledge unspoken issues with the current treatment.

If applicable, we recommend that you deposit your laboratory protocols in protocols.io to enhance the reproducibility of your results. Protocols.io assigns your protocol its own identifier (DOI) so that it can be cited independently in the future. For instructions see: http://journals.plos.org/plosone/s/submission-guidelines#loc-laboratory-protocols . Additionally, PLOS ONE offers an option for publishing peer-reviewed Lab Protocol articles, which describe protocols hosted on protocols.io. Read more information on sharing protocols at https://plos.org/protocols?utm_medium=editorial-email&utm_source=authorletters&utm_campaign=protocols .

We look forward to receiving your revised manuscript.

Kind regards,

Ryan K. Roeder, PhD

Academic Editor

PLOS ONE

Journal Requirements:

Reviewers' comments:

Reviewer's Responses to Questions

**Comments to the Author**

1. If the authors have adequately addressed your comments raised in a previous round of review and you feel that this manuscript is now acceptable for publication, you may indicate that here to bypass the “Comments to the Author” section, enter your conflict of interest statement in the “Confidential to Editor” section, and submit your "Accept" recommendation.

Reviewer #1: (No Response)

Reviewer #3: (No Response)

2. Is the manuscript technically sound, and do the data support the conclusions?

Reviewer #1: Partly

Reviewer #3: Yes

3. Has the statistical analysis been performed appropriately and rigorously? 

Reviewer #1: I Don't Know

Reviewer #3: No

4. Have the authors made all data underlying the findings in their manuscript fully available?

Reviewer #1: Yes

Reviewer #3: Yes

5. Is the manuscript presented in an intelligible fashion and written in standard English?

Reviewer #1: Yes

Reviewer #3: Yes

6. Review Comments to the Author

Reviewer #1: (No Response)

Reviewer #3: This manuscript presents a framework for increasing effective

detector resolution via spatially varying filtering for an encoding

component in the forward model of X-ray CT systems. It is primarily

a proposal for a family of techniques to match the dimension of

the measurements from augmented detectors to the unknown

line integrals of basis materials.

The approach is creative and the simulated, preliminary results

are encouraging. There are improvements that would strengthen

the paper, and this will be worthy of publication with what I would

term minor revisions.

Primary points of concern:

1. Section 2.6 on Covariance Analysis should be more fully explained.

Given the form of the forward model, the Jacobian would appear to be

a function of the unknown parameters and nontrivial to compute. Was

the Jacobian made available by the Matlab function cited? I’d like to

see that parameterized matrix for the 4x4 example, but at the least we

should know how it’s computed and plugged into (8) for the given case.

Also, given the nonlinearity of the relationships, (8) should be given as an approximation

rather than equality. Probably the best we could do here would be an inequality

relative to the Cramer-Rao lower bound, which would require

an expectation over the data space for each Jacobian entry.

Especially since there would be a number of ways to increase the dimension

of the measurement space, we should also consider the case in which the

Jacobian is not a square matrix.

2. The covariances are presented with little comment beyond noting the

several negative entries. The 4x4 matrix has a relatively large (>10^4)

condition number for a small matrix, raising questions about the information,

with regard to the parameters, available in the 4 measurements. The results

in Figures 3-6 are encouraging, but some discussion of their potential sensitivity

to modeling errors and noise should appear in light of the eigenstructure

of the given, approximate covariance. We must keep in mind as well that mean-squared error

in an estimator includes both variance and bias^2. Further, the ill effects of

variances depend on the signal strength (~mean) of the true values. In the

example used, the approximated variances are high for elements 1 and 2 of the parameter

vector, but I believe these are both associated with the water component, which has a

much higher mean value than the bone in the case shown.

3. With the simulations already programmed, it should not be a large effort

to vary the filtration in the “medical” case to see whether the information matrix

might be equalized a bit to provide better estimation of the four components.

Just working through the same type of approximate covariance structure would suffice.

It is mentioned in the Discussion that filter optimization will be the subject of future work,

but one perturbation would be nice to see here, too, and it could be fully explored

in the next paper.

A few details:

1. Line 31: “spectrum” for singular

2. Line 37: “the and”

3. Line 210: “specta”

4. Line 215: don’t need to capitalize “titanium”.

5. Line 240: adding noise to “raw detector measurements”, or to noiseless values?

6. Line 298 and following: “Albeit” appears at least four times in less than two pages,

which is a bit jarring. Also, I’ve never before seen a sentence begin with this word.

May want to check a usage reference for this one.

7. PLOS authors have the option to publish the peer review history of their article (what does this mean?). If published, this will include your full peer review and any attached files.

Reviewer #1: No

Reviewer #3: No

---

## [Editor Report · Decision Letter 2]

29 Sep 2021

Sub pixel resolution using spectral-spatial encoding in x-ray imaging

PONE-D-21-12689R2

Dear Dr. Szczykutowicz,

We’re pleased to inform you that your manuscript has been judged scientifically suitable for publication and will be formally accepted for publication once it meets all outstanding technical requirements.

Kind regards,

Ryan K. Roeder, PhD

Academic Editor

PLOS ONE

Additional Editor Comments (optional):

Please note an incomplete sentence on p16 of the revised manuscript.  This can be addressed during page proofs.

---

## [Editor Report · Acceptance letter]

12 Oct 2021

PONE-D-21-12689R2

Sub pixel resolution using spectral-spatial encoding in x-ray imaging

Dear Dr. Szczykutowicz:

I'm pleased to inform you that your manuscript has been deemed suitable for publication in PLOS ONE. Congratulations! Your manuscript is now with our production department.

Kind regards,

on behalf of

Dr. Ryan K. Roeder

Academic Editor

PLOS ONE